# Vaccination Strategies against Seasonal Influenza in Long Term Care Setting: Lessons from a Mathematical Modelling Study

**DOI:** 10.3390/vaccines11010032

**Published:** 2022-12-23

**Authors:** Matteo Ratti, Diego Concina, Maurizio Rinaldi, Ernesto Salinelli, Agnese Maria Di Brisco, Daniela Ferrante, Alessandro Volpe, Massimiliano Panella

**Affiliations:** 1Department of Translational Medicine (DiMeT), Università del Piemonte Orientale, 28100 Novara, Italy; 2Department of Pharmaceutical Science (DSF), Università del Piemonte Orientale, 28100 Novara, Italy; 3Department of Studies for Economics and Business (DiSEI), Università del Piemonte Orientale, 28100 Novara, Italy

**Keywords:** mathematical modelling of infectious diseases, influenza, long term care facilities, health care workers

## Abstract

Background: seasonal influenza in nursing homes is a major public health concern, since in EU 43,000 long term care (LTC) facilities host an estimated 2.9 million elderly residents. Despite specific vaccination campaigns, many outbreaks in such institutions are occasionally reported. We explored the dynamics of seasonal influenza starting from real data collected from a nursing home located in Italy and a mathematical model. Our aim was to identify the best vaccination strategy to minimize cases (and subsequent complications) among the guests. Materials and methods: after producing the contact matrices with surveys of both the health care workers (HCW) and the guests, we developed a mathematical model of the disease. The model consists of a classical SEIR part describing the spreading of the influenza in the general population and a stochastic agent based model that formalizes the dynamics of the disease inside the institution. After a model fit of a baseline scenario, we explored the impact of varying the HCW and guests parameters (vaccine uptake and vaccine efficacy) on the guest attack rates (AR) of the nursing home. Results: the aggregate AR of influenza like illness in the nursing home was 36.4% (ward1 = 56%, ward2 = 33.3%, ward3 = 31.7%, ward4 = 34.5%). The model fit to data returned a probability of infection of the causal contact of 0.3 and of the shift change contact of 0.2. We noticed no decreasing or increasing AR trend when varying the HCW vaccine uptake and efficacy parameters, whereas the increase in both guests vaccine efficacy and uptake parameter was accompanied by a slight decrease in AR of all the wards of the LTC facility. Conclusion: from our findings we can conclude that a nursing home is still an environment at high risk of influenza transmission but the shift change room and the handover situation carry no higher relative risk. Therefore, additional preventive measures in this circumstance may be unnecessary. In a closed environment such as a LTC facility, the vaccination of guests, rather than HCWs, may still represent the cornerstone of an effective preventive strategy. Finally, we think that the extensive inclusion of real life data into mathematical models is promising and may represent a starting point for further applications of this methodology.

## 1. Introduction

According to recent studies, seasonal influenza still represents a major threat for the aged population [1,2]. In fact, the European Centre for Disease Prevention and Control (ECDC) estimates that in the European Union (EU) every year this pathogen is responsible of 4–50 million of symptomatic infections, which in turn lead to 15,000–70,000 deaths. It has been recently calculated that 88% of those occured among people 65 years and older and that the rates of mortality from influenza complications in this age group were roughly 35 times higher compared with those <65 years [3].

At present time, in EU 43,000 long term care (LTC) facilities host an estimated 2.9 million elderly residents [4]. Therefore, the impact of seasonal influenza towards those institutions represent a major public health concern. Nursing homes are structures where an outbreak of seasonal influenza could lead to serious consequences, especially for frail residents [5]. Moreover, it is now known that such outbreaks can occur even in vaccinated populations because of the many different factors which have an impact on the vaccine effectiveness [6,7,8]. For example, the vaccine matching with current circulating strains and the immunosenescence of the older patient are just two important challenges for the current research [9,10]. In fact, influenza outbreaks in LTC facilities are occasionally reported in literature from many countries [11,12], despite the worldwide specific vaccination campaigns: as an example, Mahmud and coll. [13] reported the paradigmatic example of 154 ILI outbreaks in Canadian nursing homes where on average 92% of the guests were vaccinated.

In this context it is not yet clear what the best vaccination strategy may be in order to contain or reduce attack rates (AR) of influenza among guests. Even though the World Health Organization [14] and some studies [15,16] provided a recommendation in favour of HCW vaccination, a Cochrane systematic review of cluster Randomized Controlled Trials (cRCT) and cohort studies by Thomas and coll. found no conclusive evidence supporting programs for HCW vaccination [17]. Therefore, a knowledge gap currently exists between what is recommended and what may be effective both in terms of burden of disease and economic cost.

Since cRCT are becoming less feasible due to ethical reasons, a significant contribution in understanding the dynamics of the infection in such institutions come from the mathematical modelling of infectious diseases. Some authors already applied this methodology to influenza spreading in nursing homes and hospital departments and concluded that an increase in health care workers (HCW) vaccination uptake may lead to lower ARs for guests [18,19,20]. However, their studies were prominently theoretical and not included in the Cochrane review, probably because were reputed not to be capable of providing high quality evidence.

Starting from real data collected during the 2019/2020 season in a LTC facility located in Biella (Italy), we investigated the dynamic of seasonal influenza with a mathematical model. Our aim was to identify the best vaccination strategy in the nursing home to minimize cases (and subsequent complications) among the guests exploring different levels of vaccination uptake and efficacy of both guests and HCWs. In the following paragraphs, we first describe the 2019/2020 influenza season in Italy and in a local nursing home. Next, we explain the development of a mathematical model composed of a classical deterministic SEIR model for the general population and a stochastic agent based model that formalize the dynamics of the influenza inside the nursing home. After a model fit to assess the correct probabilities of transmission, we provide the results of varying the HCW and Guest vaccination uptake and efficacy parameters. The originality of our approach consists in the extensive incorporation of real data into the mathematical model.

## 2. Materials and Methods

We developed a mathematical model to describe the spreading of influenza in a long-term care nursing home, extending the one introduced by van den Dool and coll [18]. We developed it and investigated its fitting capabilities by using data from a real LTC facility located in Biella, Piedmont, Northern Italy, during the season 2019/2020, prior to analyzing the impact of scenarios of different vaccine uptake/efficacy parameters among guests and HCWs. When we analyze or compare real data, we consider influenza like illness (ILI) cases as a proxy of influenza cases because of the impossibility of a laboratory certainty diagnosis (see Section 2.5 later).

### 2.1. The Context

#### 2.1.1. The 2019/2020 Influenza Season

We considered the influenza season 2019/2020 starting from week 42 of 2019 to week 17 of 2020 (196 days) because this is the time span of the national ministerial surveillance program INFLUNET [21], during which it provides a weekly report. According to a recent study the total attack rate in Italy of influenza in this season was estimated to be circa 20% [22]. Figure 1 represents the dynamics of influenza infection in Italy during the studied season (black line) along with the 9 previous ones. Based on the epidemiological characteristics, the ministerial superior health institute (ISS) every year provides a judgement of the epidemic among the categories high/medium/low intensity. The 2019/2020 season was judged to be of “medium intensity”. Although the seasonal influenza epidemic shows some variability through the years, we reasonably think that the characteristics of the 2019/2020 season are roughly in line with the majority of the previous ones. In fact, Trentini and coll [22] reported that in the decade ending with the 2019/2020 season the total influenza AR ranged from 12.7% in the 2016–2017 season to 30.5% in the 2017–2018 season, with 6 out of 10 AR estimates of circa 20%. The regional office of Piedmont (responsible of the vaccine supply on behalf of the national health service) provided only the standard adjuvanted quadrivalent vaccine for the 2019/2020 season (i.e., no high dose). Even if in other developed countries such as the USA there is evidence that the burden of influenza during this season was higher than the previous ones [23], in Italy the intensity was judged “medium” like the majority of previous ones.

#### 2.1.2. The Long Term Care Nursing Home

The LTC facility “Belletti Bona” is located in Biella, Piedmont, Northern Italy. It has 120 beds, divided into 4 wards: w1, w2, w3 and w4. The w1 ward consists of 20 beds for self-sufficient guests, who participate into daily recreational activities. Ward w2 has 20 beds reserved for guests who are affected by severe chronic diseases such as dementia, and is isolated from the rest of the facility. Ward 3 and 4 include 40 beds each and their guests need high intensity assistance for common comorbidities of the elderly. All the rooms of the facility have 2 beds. Further details about the characteristics of the nursing home are provided in Appendix A.

### 2.2. Mathematical Model

Our model is structured with a classical SEIR (susceptible, exposed, infected and removed) deterministic model describing the influenza spreading in the general population and a stochastic agent based model that formalizes the dynamics of the infection inside the nursing home. We provide here a brief description of the model along with its key characteristics and parameters, leaving a more extensive explanation in Appendix A. In particular, the mathematical equations describing the change of state of the individuals during the time period considered are reported in Appendix A. The model was implemented in the programming language R [24].

#### 2.2.1. Population Model

For every individual entering the nursing home the population model calculates a probability of belonging to each of the SEIR states so that during a simulation she/he is assigned to one of the compartments accordingly. This model is time continuous and utilizes three age classes: children (0–14), adults (15–64) and elderlies (over 65). We assumed no demography nor age class transitions in the time period considered. The equations describing the model state changes are reported in Appendix A.

#### 2.2.2. Stochastic Model

The stochastic model calculates for every time unit the SEIR state of every individual in the LTC facility. This calculation is made on the basis of the contact pattern of each person along with the likelihood that their contacts lead to an infection. For this to be possible, the model needs the contact pattern of each agent. Therefore, we developed the contact matrices from a survey of the guests (or their care givers if the guest was affected by a cognitive impairment) and HCWs during two weeks of ordinary working conditions. For this model we adopted a discrete-time approach, by setting the time step to 8 h according to the three working shifts in the structure: morning shift M (6 am to 2 pm), afternoon shift A (2 pm to 10 pm) and night shift N (10 pm to 6 am). We separately modeled HCWs, nurses, and other workers because of their different contact and working characteristics. For instance, there were some workers categories that had no close contacts with the guests (such as the psychologists). The detailed description of the different categories along with their contact matrices are reported in Appendix A. The equations describing the state changes are reported in Appendix A.

### 2.3. Key Parameters: Baseline Scenario

We defined a baseline scenario with the parameters reported in Table 1. We assumed that all HCWs belonged to adult age class (15–64) and all guests to the elderly one (over 65). HCW vaccine uptake (5.1%) was taken from the corresponding age class general population vaccination uptake, thereby assuming that being an HCW did not change the probability of vaccination. This datum has also been verified by comparison to two local studies about HCW vaccination uptake in LTC setting, which yielded comparable results of 3% [25] and 9.6% [26]. We derived the guest vaccination uptake (46.5%) from a survey of the nursing home occupants. This value is less than the corresponding age class uptake of the general population (54.6% in Italy, 51% in Piedmont region) [27], reflecting a higher hesitancy of the guests towards vaccination. We derived the vaccination effectiveness baseline parameter both for HCWs and for guests from a recent systematic review of test negative design studies [28]. The considered vaccine parameters were relative to the standard adjuvanted quadrivalent vaccine (i.e., not the high dose one), because it was the only typology of vaccine provided by the national health service at the time. The initial removed fraction of the HCWs and guests were extracted from three studies about influenza dynamics [29,30,31].

A key parameter in influenza transmission dynamics is represented by the likelihood that a certain typology of contact leads to an infection. We distinguished three cases of contacts each of those with its own probability of producing an infection: p_casual for casual contacts, p_close for close contacts, and p_change for the specific contact happening in shift-change rooms. We defined a casual contact a verbal interaction between two individuals, whereas the contact was considered close if it involved physical procedures. Consistently with other similar modelling studies [18,20], we considered the ratio p_close/p_casual fixed to 2, reflecting the fact that the physical contact carries a higher risk of infection in comparison to a verbal interaction. However, as a sensitivity analysis we explored the behaviour of the model with other values of this ratio (1.5 and 2.5).

#### Model Fit

To properly identify the probability parameters, we conducted a model fit with the baseline scenario parameters described above: the combination of these parameters that minimized the Root Mean Square Error (RMSE) of the AR of the whole structure compared to the real AR of the 2019/2020 season was selected for the subsequent analysis. We explored the following values of p_casual: 0.15, 0.20, 0.25, 0.30 along with the values of p_change: 0.05, 0.1, 0.15, 0.20, 0.25, 0.30, 0.35. Therefore, the total combinations of parameters values explored were 28; for each of them we planned 200 stochastic simulations.

### 2.4. Simulation Plan

For every combination of parameters (scenario), we planned to perform 200 replicates. We then calculated the mean AR of each scenario along with its 95% confidence interval value.

### 2.5. Outcome and Data Analysis

Consistently with the literature about influenza dynamics modelling, we selected as outcome the guest Attack Rate of the facility, calculated as the number of total infected guests along the season divided by the number of the present guests (at risk). We also calculated this measure for each of the wards. An alternative definition of AR is cumulative incidence, or incidence proportion. As a proxy for influenza cases, we registered the Influenza Like Illness (ILI) cases as defined by INFLUNET and ECDC influenza global surveillance program. In detail, a guest showing a sudden onset and at least one of the principal four systemic symptoms (fever or feverishness (T° > 38 °C), malaise, headache, myalgia) accompanied with at least one of the main respiratory symptoms (cough, sore throat, shortness of breath) was categorized as an ILI case. All the data was analyzed with R software ver. 4.1.0 (18 May 2021) [24] and RStudio ver.1.4.1717 [32].

## 3. Results

### 3.1. The 2019/2020 Influenza Season in the “Belletti Bona” LTC Nursing Home

During the season, ward 1 hosted 25 individuals, the second 27, whereas the last two 63 and 58 respectively, for a total of 173 guests. Out of a total of 63 ILI cases, 14 belonged to ward 1, 9 to ward 2, and 20 cases were registered in each of the last two wards (w3 and w4). The aggregate AR of ILI in the nursing home during the 2019/2020 season was 36.4%. The highest AR was observed in ward 1 (56%), followed by ward 4 (34.5%) and ward 2 (33.3%). The least AR of 31.7% was relative to ward 3. Figure 2 shows the AR dynamics (cumulative incidence) of the nursing home during the study period. During the study period no COVID-19 cases were registered among the guests.

### 3.2. Model Fit

The total number of simulations performed for the model fit were 5600. Among all the explored values of p_casual and p_change, the configuration of parameters that minimized the whole structure root mean square error (RMSE) is p_casual = 0.3, p_close = 0.6, p_change = 0.2. The corresponding RMSE is 0.078. Figure 3 shows an heatmap of the explored values of p_casual and p_change. The values of p_close were fixed at double the values of p_casual. Moreover, we performed a sensitivity analysis with varying levels of the p_close/p_casual ratio (Appendix A), which produced similar results. The results of the model fit for each ward are illustrated in Appendix A.

### 3.3. Further Simulations

We performed 200 replicates for each combination of parameters values (784). The final number of simulations was 156,800. Table 2 reports the mean AR values (along with their 95% CI) of both the vaccine efficacy and uptake parameters for HCWs, whereas Table 3 shows the results of the same parameters for Guests.

We were unable to find any significant trend of the mean ARs in any of the wards both when varying HCW vaccine efficacy and HCW vaccine uptake parameters (Table 2). Regarding the Guest parameters (Table 3), our results show a small trend towards mean ARs reduction when exploring increasing values of both vaccine efficacy and vaccine uptake.

For the different levels of HCW vaccine efficacy the observed mean ARs in both ward 1 and 2 range from 57% to 60%, in ward 3 from 35% to 38% and in ward 4 from 35% to 36%. The baseline scenario mean AR values for the wards were respectively: 59%, 59%, 37% and 36%. HCW vaccine uptake different levels produced mean AR ranges of 56% to 59% in ward 1, 58% to 60% for ward 2, 35% to 38% in ward 3 and 35% to 36% in ward 4. The baseline scenario mean AR values for the wards were respectively: 59%, 59%, 37% and 36%.

Considering the different levels of Guest vaccine efficacy, the observed mean ARs in ward 1 ranges from 52% to 61%, in ward 2 from 51% to 61%, in ward 3 from 32% to 38% and in ward 4 from 32% to 37%. The baseline scenario mean AR values for the wards were respectively: 59%, 59%, 37% and 36%. Guests vaccine uptake different levels produced AR ranges of 52% to 64% in ward 1, 52% to 66% for ward 2, 33% to 39% in ward 3 and 32% to 38% in ward 4. The baseline scenario AR values for the wards were respectively: 59%, 59%, 37% and 36Ṫhe simulations investigating different R0 levels (Initial Removed fraction) of both HCW and guests produced similar results. They are reported in Appendix A.

Considering the different levels of Guest vaccine efficacy, the observed mean ARs in ward 1 ranges from 52% to 61%, in ward 2 from 51% to 61%, in ward 3 from 32% to 38% and in ward 4 from 32% to 37%. The baseline scenario mean AR values for the wards were respectively: 59%, 59%, 37% and 36%. Guests vaccine uptake different levels produced AR ranges of 52% to 64% in ward 1, 52% to 66% for ward 2, 33% to 39% in ward 3 and 32% to 38% in ward 4. The baseline scenario AR values for the wards were respectively: 59%, 59%, 37% and 36Ṫhe simulations investigating different R0 levels (Initial Removed fraction) of both HCW and guests produced similar results. They are reported in Appendix A.

## 4. Discussion

Our aim was to explore vaccination strategies against influenza in a nursing home and obtain some insights about what the best strategy may be in order to contain or lower the AR among guests. The 2019/2020 season ILI attack rate in the nursing home was found to be 36.4%, considerably higher in respect to the estimates of a recent study by Trentini and coll [22], who calculated for this season an influenza mean total attack rate of 20%. The official ending INFLUNET report (relative to week 17 of 2020) estimated 7,594,700 ILI cases [33] for the 2019/2020 season (out of a population of 59,641,488 residents [34]) in Italy, for an AR of 12.73%. Even if we do not expect a nursing home guest to have the same risk of general population, we think that this difference is wide enough to categorize the nursing home as an environment of high risk for influenza disease, even considering that the facility had employed and currently employs all the prescribed an up-to-date preventive measures. Apart from the first ward, with an AR of 56%, the others showed similar and much lower ARs. This aspect may be due to the different characteristics of the population in ward 1, composed by self-sufficient guests who occasionally participated into daily recreational activities and therefore had many contacts with other individuals. On the contrary, the health status of the elderlies in the wards w2,w3 and w4 forced them to remain in bed or in their room for most of the day. As a consequence, they were less exposed to any potential source of infection. We think that the circulating Sars-CoV-2 virus did not significantly affect our results, since there have been no cases nor fatalities in the nursing home from COVID-19 during the first wave of the pandemic.

This increased risk of the whole structure in comparison to other situations is reflected in our model fit, which yielded a probability of transmission of the casual contact of 0.3. To our knowledge, this value is the highest found in the modelling literature for such a typology of contact. For instance, van den Dool and coll. [18] reported values ranging from 0.1 to 0.16 for such a contact typology, whereas a recent study about influenza estimated a household contact probability of transmission of 0.17 in a school setting [35]. Interestingly, the model fit returned a probability of transmission for the specific contact between HCWs at the moment of the shift change of 0.2, which is much less than the probability of the casual contact for the entire structure. This could mean that no further precautions (such as Personal Protective Equipment, for instance) are needed in addition to the standard protocol at the moment of the handover (or in the shift change room). In other words, the moment and the place of handover could carry no increased risk for influenza spreading in the nursing home.

The results of the simulations with varying levels of HCWs uptake and vaccine efficacy produced no significant change in the ARs of guests. This finding is consistent with the Cochrane systematic review of Thomas and coll. [17], who concluded that HCW vaccination resulted in a nil risk difference for laboratory proven influenza cases among guests in comparison to facilities where HCW vaccination was practiced. Remarkably, the studies included in the Cochrane review were cRCT or observational studies; no mathematical modelling studies were taken into account. Our results are in contrast with the findings of other mathematical modelling studies such as the ones by van den Dool [18,19] and Wendelboe [20], who observed the reduction of influenza cases when increasing the proportion of vaccinated HCWs. Nevertheless, our study differs from them by using data from a real nursing home to build the model, so it seems that by extensively incorporating data from real situation and model fit we could obtain results similar to the ones obtained from cRCTs, which are known to provide high quality evidence. This could suggest that the methodology of mathematical modelling might produce high quality evidence as long as real data is taken into account. We also think that this discrepancy could at least in part be imputed to the fact that we modeled HCWs separately from others workers such as nurses, physiotherapists and psychologists. We thought that the differences in contact patterns between other workers and, most of all, the guests deserved a specific consideration in the model. In fact, including such aspect may provide different results (and therefore different recommendations or preventive measures) depending on the typology of worker. A possible explanation for this finding could be that a single infected visitor or HCW may act as a carrier among guests, so that, unless we can obtain values close to 100% HCW uptake and to 100% vaccine efficacy, the disease has sufficient transmitting potential. This may happen because there are enough individuals either unvaccinated or non-responders, and therefore can help the spreading of the disease. However, we did not explore such extreme parameters because unrealistic to achieve in practice.

As expected, the increase in both vaccine uptake and efficacy parameters of guests (and the initial removed fraction of guests) resulted in a slight decrease of their ARs.Therefore, we may deduce that the vaccination of the guest still plays a major role in reducing ARs and should still be the cornerstone of a successful vaccination campaign in LTC setting. In other words, our results strengthen the widespread recommendation for the vaccination of LTC guests. However, these results were accompanied with very large 95% confidence intervals, revealing that the number of simulations were probably insufficient for a strong assertion. For the future modelling studies that we planned, we expect to conduct a preliminary power analysis with a higher number of simulation and an appropriate hardware, which represented a limitation in our study because of its scarce availability and high cost.

Differently from ward 1,3, and 4, the model produced mean ARs for ward 2 that did not adhere to the real life data. In fact, their ranges were 57–60% and 58–60% when varying HCW parameters; 51–61% and 52–66% when varying Guest parameters. The actual w2 AR for the studied season was instead 33.3%. We expected the calculated mean ARs of ward 2 to be much more similar to the ones of ward 3 and 4, since the population of those wards (who needed high assistance) behaved in a similar manner, at least in theory. However, a significant number of w2 guests were affected by cognitive impairment, so the contact pattern was often indirectly obtained by the survey of their care-givers rather than themselves. We think that this factor distorted the structure of the contact pattern. We are therefore cautious in interpreting the results of ward 2 and we are considering, for future studies, to exclude wards where a significant part of the guests are affected by cognitive impairment.

### Strengths and Limitations

Our model is built upon data collected from a real nursing home and a model fit. To our knowledge, this is the first attempt to include in a mathematical model for influenza spreading peculiar details such as the different contact patterns of health care workers, nurses, and other workers, or the modelling of the handover circumstances.

Even though we believe that the characteristics of the nursing home are roughly typical, our results and their interpretation may not be generalizable to facilities or situations different from the one in question. For example, there are many smaller nursing homes to which our deductions may not be applicable. Second, we point out that the study was conducted considering ILI cases, rather than laboratory confirmed influenza. Recent estimates provide that influenza virus may represent 12.5–55% of ILIs [36,37]. ILI cases is a measure that may overestimate influenza cases because of other similar respiratory infectious diseases. However, the majority of other ILI-related pathogens share similar epidemiological characteristics to influenza: for instance, Respiratory Syncytial Virus, Adenovirus, Rhinovirus have the same airborne transmission dynamics and represent the most common causes of ILI [38]. Also, Sars-CoV-2 virus may have represented a common etiology of ILI in the considered period. However, given that the national health service provided swabs to nursing homes as a priority and the fact that no COVID cases were found in the nursing home during the first wave, we reasonably think that the lesson learned from this and other studies concerning ILIs are also valid for influenza.

Nevertheless, until further seasons will be investigated we recommend caution in generalizing our results, because we can not ensure that the intensity and the burden of disease during the 2019/2020 season were atypical despite the ministerial judgment of “medium” intensity. Noteworthy, common comorbidities of the elderlies that implied a different contact pattern HCW-Guest were not modeled in this study. As an example, guests affected by COPD or other lung diseases may require closer and more frequent assistance than an ordinary guest. Moreover, they could be more susceptible to respiratory infections. For future works we will take this factor into account.

Another limitation of our study regarded the absence of a specific vaccination class in our model, so that the Removed state included both immunity derived from prior infections and from vaccination. We are therefore unable to discriminate the reason at the basis of the immunity of a single agent but this was not a research question.

Finally, the number of simulations in our study appeared to be insufficient for strong assertions. For future similar studies, we recommend a preliminary power analysis.

## 5. Conclusions

From our findings we can conclude that a nursing home is still an environment at high risk of influenza transmission but the shift change room and the handover situation carry no higher relative risk. Therefore, additional preventive measures in this circumstance may be unnecessary. In such a closed environment, the vaccination of guests, rather than HCWs, may still represent the cornerstone of an effective preventive strategy. We already have planned a study with a higher number of simulations, exploring the impact of the timing of the vaccination campaign and how the epidemiological characteristics of the influenza season may condition the ARs inside the nursing home. Finally, we think that the extensive inclusion of real life data into mathematical models is promising and may represent a starting point for further applications of this methodology.

## Figures and Tables

**Figure 1 vaccines-11-00032-f001:**
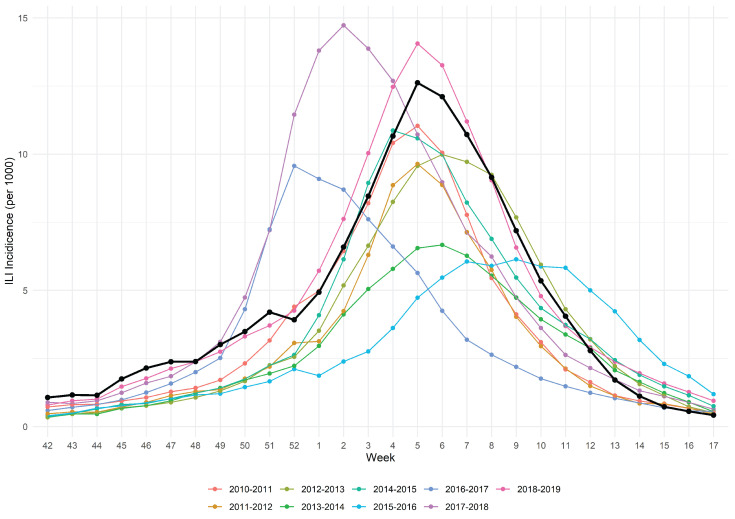
ILI incidence of 2019/2020 influenza epidemic (black line) along with 9 previous seasons.

**Figure 2 vaccines-11-00032-f002:**
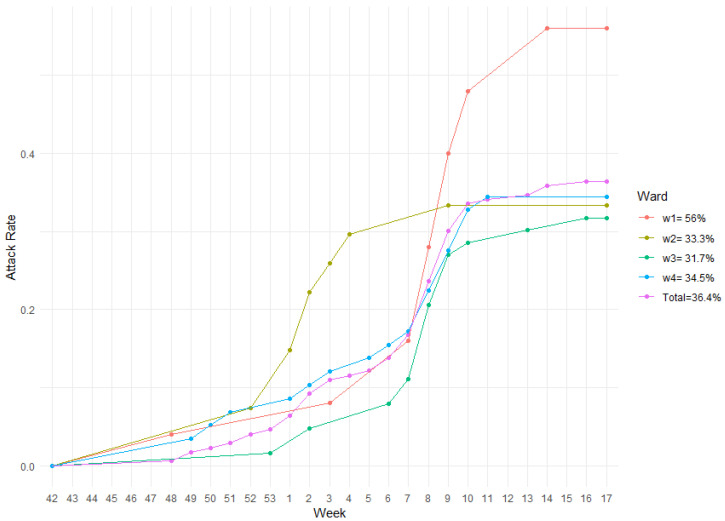
AR of the whole facility along with its wards during influenza season 2019/2020.

**Figure 3 vaccines-11-00032-f003:**
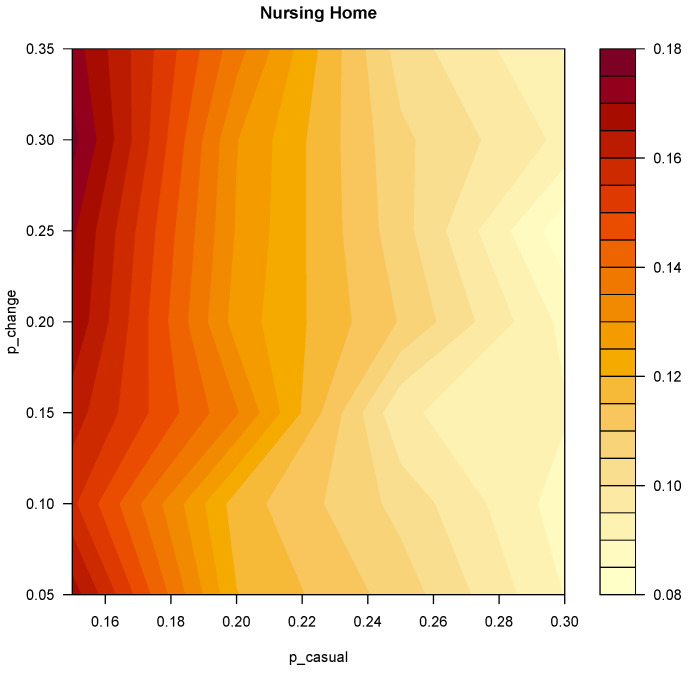
Heatmap representing the RMSE of the model fit for the whole facility. On the X and Y axes the explored values of p_casual and p_change respectively.

**Table 1 vaccines-11-00032-t001:** Summary of the main parameters of the mathematical model.

Parameter	Baseline Scenario Values	Reference	Values Explored with Further Simulations
Casual contact probability of transmission—p_casual			
(unity: contact^−1^)	model fit	-	0.3
Close contact probability of transmission—p_close			
(unity: contact^−1^)	model fit	-	0.6
Shift change probability of transmission—p_change			
(unity: contact^−1^)	model fit	-	0.2
Initial Removed Fraction of HCWs	0.10	[29,30,31]	0-0.05-0.10-0.2-0.3
Initial Removed Fraction of Guests	0.20	[29,30,31]	0-0.05-0.10-0.2-0.3
Vaccine effectiveness on HCWs	0.3744	[28]	0.1-0.2-0.37-0.5-0.6
Vaccine effectiveness on Guests	0.21	[28]	0.1-0.2-0.3-0.4
Guests Vaccine Uptake	0.4653	derived from LTC facility real data	0-0.25-0.46-0.5-0.75-1
HCW Vaccine Uptake	0.0516	[27]	0-0.05-0.25-0.5-0.75-1

**Table 2 vaccines-11-00032-t002:** Results of the simulations: HCW vaccine efficacy and vaccine uptake parameters. The values belonging to the baseline scenario are reported in bold.

HCW Vaccine Efficacy	Ward	AR (Mean)	LCI 95%	UCI 95%	HCW Vaccine Uptake	Ward	AR (Mean)	LCI 95%	UCI 95%
0.1	1	0.57	0.05	0.81	0.00	1	0.58	0.37	0.81
0.2	1	0.59	0.4	0.8	**0.05**	**1**	**0.59**	**0.38**	**0.81**
**0.37**	**1**	**0.59**	**0.38**	**0.81**	0.25	1	0.59	0.38	0.81
0.5	1	0.58	0.36	0.81	0.50	1	0.56	0.05	0.77
0.6	1	0.6	0.37	0.82	0.75	1	0.59	0.36	0.77
					1.00	1	0.56	0.30	0.77
0.1	2	0.59	0.39	0.82	0.00	2	0.58	0.32	0.81
0.2	2	0.57	0.27	0.77	**0.05**	**2**	**0.59**	**0.31**	**0.81**
**0.37**	**2**	**0.59**	**0.31**	**0.81**	0.25	2	0.60	0.36	0.81
0.5	2	0.58	0.04	0.85	0.50	2	0.58	0.07	0.78
0.6	2	0.6	0.36	0.81	0.75	2	0.59	0.35	0.80
					1.00	2	0.59	0.29	0.81
0.1	3	0.37	0.21	0.53	0.00	3	0.36	0.22	0.50
0.2	3	0.38	0.23	0.5	**0.05**	**3**	**0.37**	**0.22**	**0.50**
**0.37**	**3**	**0.37**	**0.22**	**0.5**	0.25	3	0.37	0.22	0.53
0.5	3	0.36	0.22	0.5	0.50	3	0.37	0.21	0.56
0.6	3	0.35	0.17	0.49	0.75	3	0.38	0.21	0.54
					1.00	3	0.35	0.21	0.50
0.1	4	0.35	0.21	0.52	0.00	4	0.36	0.22	0.50
0.2	4	0.36	0.24	0.51	**0.05**	**4**	**0.36**	**0.17**	**0.52**
**0.37**	**4**	**0.36**	**0.17**	**0.52**	0.25	4	0.35	0.22	0.51
0.5	4	0.36	0.21	0.54	0.50	4	0.36	0.18	0.51
0.6	4	0.36	0.21	0.51	0.75	4	0.35	0.14	0.50
					1.00	4	0.35	0.20	0.50

**Table 3 vaccines-11-00032-t003:** Results of the simulations: Guests vaccine efficacy and vaccine uptake parameters. The values belonging to the baseline scenario are reported in bold.

Guests Vaccine Efficacy	Ward	AR (Mean)	LCI 95%	UCI 95%	Guests Vaccine Uptake	Ward	AR (Mean)	LCI 95%	UCI 95%
0.1	1	0.61	0.36	0.85	0.00	1	0.64	0.42	0.90
**0.2**	**1**	**0.59**	**0.38**	**0.81**	0.25	1	0.64	0.42	0.90
0.3	1	0.52	0.06	0.71	**0.46**	**1**	**0.59**	**0.38**	**0.81**
0.4	1	0.54	0.32	0.80	0.50	1	0.60	0.38	0.85
					0.75	1	0.55	0.34	0.76
					1.00	1	0.52	0.32	0.76
0.1	2	0.61	0.05	0.85	0.00	2	0.66	0.45	0.90
**0.2**	**2**	**0.59**	**0.31**	**0.81**	0.25	2	0.60	0.05	0.82
0.3	2	0.56	0.08	0.80	**0.46**	**2**	**0.59**	**0.31**	**0.81**
0.4	2	0.51	0.09	0.70	0.50	2	0.59	0.33	0.81
					0.75	2	0.54	0.05	0.76
					1.00	2	0.52	0.20	0.77
0.1	3	0.38	0.24	0.56	0.00	3	0.39	0.22	0.57
**0.2**	**3**	**0.37**	**0.22**	**0.50**	0.25	3	0.39	0.24	0.54
0.3	3	0.35	0.18	0.49	**0.46**	**3**	**0.37**	**0.22**	**0.50**
0.4	3	0.32	0.13	0.48	0.50	3	0.38	0.24	0.51
					0.75	3	0.34	0.20	0.49
					1.00	3	0.33	0.19	0.52
0.1	4	0.37	0.05	0.54	0.00	4	0.38	0.26	0.56
**0.2**	**4**	**0.36**	**0.17**	**0.52**	0.25	4	0.37	0.02	0.57
0.3	4	0.33	0.18	0.49	**0.46**	**4**	**0.36**	**0.17**	**0.52**
0.4	4	0.32	0.21	0.47	0.50	4	0.35	0.21	0.50
					0.75	4	0.33	0.03	0.46
					1.00	4	0.32	0.21	0.43

## Data Availability

Data is available upon motivated request.

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
