# Peer review of "Vaccination Strategies against Seasonal Influenza in Long Term Care Setting: Lessons from a Mathematical Modelling Study"

_vaccines, 2022, doi:10.3390/vaccines11010032_

Round 1

Reviewer 1 Report

I think that the paper is well written and contains an interesting modelling study combining different methods which adequately describe the corresponding phenomena of disease spread inside and outside the nursing home. 

I have two main concerns about the modelling. Once, maybe an earlier year could have been a better choice to study as in the winter studied in the paper, COVID cases could already alter the observations concerning influenza. 

Another issue which possibly may have an important influence on the results is that according to the supplementary description, it is assumed that all visitors are in the middle age group (15-64 years), which seems rather unrealistic to me: grandchildren and friends living outside the nursing home are very likely visitors in my opinion. Moreover, the role of children might be especially important as due to their contact rates, they are usually among the most important spreaders of infectious diseases.

A couple of typos (e.g. build->built in line 298) could be corrected.

Author Response

Reviewer 1

I think that the paper is well written and contains an interesting modelling study combining different methods which adequately describe the corresponding phenomena of disease spread inside and outside the nursing home. 

Dear Reviewer,

We appreciate your valuable comments and thank you for the time dedicated to reading our work. We provide precise answers to your concern in the following paragraphs.

I have two main concerns about the modelling. Once, maybe an earlier year could have been a better choice to study as in the winter studied in the paper, COVID cases could already alter the observations concerning influenza. 

We thank you for your comment. We agree that an earlier year could have been a better choice; however, the real data collection process (for example the number of workers, the contact patterns etc…) was already in place at the time of the beginning of the pandemics. We acknowledge the fact that Sars-CoV-2 may have altered the cases, but we point out that no cases of COVID were found in the first wave. Also, the national health service provided the swabs to nursing homes as a priority, minimizing the risk that some early cases were misclassified. Therefore, the impact of Sars-CoV-2 on ILI cases should have been similar to other etiological sources such as rhinoviruses et al. However, we reinforced the caution in interpreting our results and the potential role of COVID in the limitation section on lines 339-347.

Another issue which possibly may have an important influence on the results is that according to the supplementary description, it is assumed that all visitors are in the middle age group (15-64 years), which seems rather unrealistic to me: grandchildren and friends living outside the nursing home are very likely visitors in my opinion. Moreover, the role of children might be especially important as due to their contact rates, they are usually among the most important spreaders of infectious diseases.

We appreciate your comment. Even if there was no official visitor register, we chose the most frequent age class of the visitors as an assumption in our model on the basis of quick interviews of workers. We partially agree with your opinion and acknowledge that other class age could have had an influence on the results, but since in the nursing home it is unlikely for a child to be an unaccompained visitor, we reasonably think that his/her contact pattern closely match or resembles the one of a middle-age-class individual. In other words, the behavior of individuals belonging to extreme age classes was thought to be similar to the class age 15-64, at least for the time spent inside the nursing home. Therefore, the influence on the overall results should have been negligible or of minor importance. We updated the supplementary material document (Text S1) section 2.4 (Visitors) with these details.

A couple of typos (e.g. build->built in line 298) could be corrected.

Thanks for your kind observation. We reviewed the manuscript and corrected the found typos.

Kind Regards,

Reviewer 2 Report

I was invited to revise the paper entitled "Vaccination Strategies against Seasonal Influenza in Long Term Care Setting: Lessons from a Mathematical Modelling Study". It described a mathematical model aimed to predict the dynamic of seasonal flu in a long term care facilities. This model was based on real world data obtained from a longtermcare facility from a Northern Italian Region. 

The topic is interesting and the paper is well written. The methodology used was strong and this paper can improve the knowledge on this field.

I have some observations:

- In Italy, elderly patients can be vaccinated with different type of vaccine: adjuvanted quadrivalent flu vaccine and high dose quadrivalent flu vaccine. Authors should consider the different efficacy of these vaccines in their model;

- Among limitations, Authors should consider that their model was based on data from season 2019/20, prior pandemic. Actually, evidence from surveillance systems from some countries shows as flu incidence dynamic was different, with a prior onset and strong impact. So Authors should consider that actually this work cannot be generalized to all seasons;

- In discussion section Authors should better clarify why HCWs high coverages was not associated to the decrease of the incidence;

- Other than cognitive impaired patients, Authors should also consider that patients with chronic lung disease (example COPD) also need close contact with HCW. In addition, they are more suscectible to respiratory infections, frequently needing O2 therapy.

Author Response

Reviewer 2

I was invited to revise the paper entitled "Vaccination Strategies against Seasonal Influenza in Long Term Care Setting: Lessons from a Mathematical Modelling Study". It described a mathematical model aimed to predict the dynamic of seasonal flu in a long term care facilities. This model was based on real world data obtained from a longtermcare facility from a Northern Italian Region. 

The topic is interesting and the paper is well written. The methodology used was strong and this paper can improve the knowledge on this field.

Dear Reviewer,

We appreciate your valuable comments and thank you for the time dedicated to reading our work. We provide precise answers to your concern in the following paragraphs.

I have some observations:

- In Italy, elderly patients can be vaccinated with different type of vaccine: adjuvanted quadrivalent flu vaccine and high dose quadrivalent flu vaccine. Authors should consider the different efficacy of these vaccines in their model;

We thank you for your suggestion and the opportunity of a clarification. The Piedmont Region in the 2019/2020 season only provided the standard adjuvanted quadrivalent flu vaccine both for HCW and Guests. In Italy the Regional offices organize a yearly auction with the details of the vaccines to be provided by the pharmaceutical companies. Even if we cannot exclude that some individuals may have chosen to buy themselves a high dose vaccine, this is not a common behavior, and it is unlikely to have significantly influenced our results. Here we provide the link to the auction text for the 2019/2020 Piedmont region.

https://www.scr.piemonte.it/sites/default/files/convenzioni/documenti/Capitolato%20tecnico_0.pdf

However, in the future studies that we have planned the possibility of different vaccines will be taken into account, in order to improve the generalizability of the results.

We incorporated this information in the manuscript in lines 100-105 and 158-160.

- Among limitations, Authors should consider that their model was based on data from season 2019/20, prior pandemic. Actually, evidence from surveillance systems from some countries shows as flu incidence dynamic was different, with a prior onset and strong impact. So Authors should consider that actually this work cannot be generalized to all seasons;

Thank you for your comment. We described in section 2.1.1 the general characteristics of the 2019/2020 influenza season in Italy and we added some further details (lines 102-105). Moreover, we explicitly included a comment in the limitation section (lines 344-348) that warns about the generalizability of the results, following your advice.

- In discussion section Authors should better clarify why HCWs high coverages was not associated to the decrease of the incidence;

Dear reviewer, thank you for your suggestion. We improved the discussion on lines 296-301 specifically addressing this question. In brief, a possible explanation could be that a single infected visitor or HCW may act as a carrier among guests, so that, unless we can obtain 100% HCW coverage and 100% Vaccine efficacy, the disease has sufficient transmitting potential. This may happen because there are enough individuals either unvaccinated or non-responders, and therefore can help the spreading of the disease. We did not explore such extreme parameters because unrealistic to achieve in practice.

- Other than cognitive impaired patients, Authors should also consider that patients with chronic lung disease (example COPD) also need close contact with HCW. In addition, they are more suscectible to respiratory infections, frequently needing O2 therapy.

We thank you for your comment and appreciate your interesting suggestion. We did not collect such details about comorbidities of the elderly like COPD or other lung diseases, but we plan to model this aspect in the next study. We included these details on lines 344-351.

Kind regards,

Reviewer 3 Report

In this manuscript, the authors use a classical SEIR (susceptible, exposed, infected, and removed) deterministic model describing influenza spreading in the general population and a stochastic agent-based model that formalizes the dynamics of the infection inside the nursing home. They use real data from the influenza season 2019/2020 starting from week 42 of 2019 to week 17 of 2020 (196 days). They conduct a model fit with the different baseline scenario parameters.

Although the manuscript is well written and the method, as well as results, seem correct, there are some points that must be taken into account before the acceptance of this work for publication.

1- The introduction is too vague. The authors must explicitly indicate their main contribution, as well as the obtained results and the organization of the manuscript.

2- For the reader's convenience, the authors must present the equations of the SEIR deterministic model and the equations of the stochastic model they use to perform all simulations.

3- In the Discussion section, the authors argue that "Our aim was to explore vaccination strategies against influenza in a nursing home and 221 obtain some insights about what the best strategy may be in order to contain or lower the 222 AR among guests", nevertheless, in the SEIR model, there is not a vaccination class. 

Thus it is important that the authors present the mathematical model used to obtain all the results presented in this work.

Author Response

Reviewer 3

In this manuscript, the authors use a classical SEIR (susceptible, exposed, infected, and removed) deterministic model describing influenza spreading in the general population and a stochastic agent-based model that formalizes the dynamics of the infection inside the nursing home. They use real data from the influenza season 2019/2020 starting from week 42 of 2019 to week 17 of 2020 (196 days). They conduct a model fit with the different baseline scenario parameters.

Dear Reviewer,

We appreciate your valuable comments and thank you for the time dedicated to reading our work. We provide precise answers to your concern in the following paragraphs.

 Although the manuscript is well written and the method, as well as results, seem correct, there are some points that must be taken into account before the acceptance of this work for publication.

1- The introduction is too vague. The authors must explicitly indicate their main contribution, as well as the obtained results and the organization of the manuscript.

We thank you for your suggestion. We improved the introduction section of the manuscript considering your valuable advice (lines 53-55 and 68-75), including some details about the knowledge gap and the originality of our approach.

2- For the reader's convenience, the authors must present the equations of the SEIR deterministic model and the equations of the stochastic model they use to perform all simulations.

Dear reviewer, thank you for your suggestion and for pointing out this lack of ours. We mistakenly omitted the references to the supplementary materials where the equations are clearly described. We did not think that the equations deserved a full explanation in the main paper because the aim of the journal is medical rather than technical. We incorporated the right references in the manuscript in lines 121-122,129-130 and 144-145. Please refer to supplementary materials – Text S1, sections number 1.2 and 2 (particularly Tables S13 and S23).

3- In the Discussion section, the authors argue that "Our aim was to explore vaccination strategies against influenza in a nursing home and 221 obtain some insights about what the best strategy may be in order to contain or lower the 222 AR among guests", nevertheless, in the SEIR model, there is not a vaccination class. 

Thank you for your comment. We did not explicitly model a vaccination class in this study, because the reason for an individual to be assigned to the Removed state were not under study. This means that the Removed compartment/state included both the immunity derived from previous infection and the immunity derived from vaccination. Given the fact that at the beginning of the study period (week 42) the vaccination campaign was already completed, and a significant number of cases was not observed in the first weeks, we reputed that this aspect did not significantly influence our results and their interpretations.

However, for our next study we are considering to include in the model a specific vaccination class and to extend the studied period, in order to clearly assess the reason at the basis of an individual removed state.

We discussed this limitation in the manuscript at lines 352-355.

Thus it is important that the authors present the mathematical model used to obtain all the results presented in this work.

We thank you for your suggestion. The mathematical model is described in detail in the supplementary material – Text S1. We included references to the supplementary material in the main draft, when needed.

Kind Regards,

Round 2

Reviewer 2 Report

Authors addressed all points raised during my revision. The paper can be accepted for publication.

Reviewer 3 Report

This version of the manuscript is well improved. I recommend it for publication.